# Fundamental Characterization of Antibody Fusion-Single-Chain TNF Recombinant Proteins Directed against Costimulatory TNF Receptors Expressed by T-Lymphocytes

**DOI:** 10.3390/cells12121596

**Published:** 2023-06-09

**Authors:** Hodaka Nagai, Mitsuki Azuma, Ayaka Sato, Nagito Shibui, Sayaka Ogawara, Yuta Tsutsui, Ayano Suzuki, Tomomi Wakaizumi, Aya Ito, Shimpei Matsuyama, Masashi Morita, Mari Hikosaka Kuniishi, Naoto Ishii, Takanori So

**Affiliations:** 1Laboratory of Molecular Cell Biology, Graduate School of Medicine and Pharmaceutical Sciences, University of Toyama, Toyama 930-0194, Japan; 2Department of Microbiology and Immunology, Tohoku University Graduate School of Medicine, Tohoku University, Sendai 980-8575, Japan

**Keywords:** 4-1BB, CD27, costimulation, GITR, OX40, T cell, TNFRSF, TNFSF, cytokine, cytokine receptor

## Abstract

The costimulatory signal regulated by the members of the tumor necrosis factor receptor (TNFR) superfamily expressed by T cells plays essential roles for T cell responses and has emerged as a promising target for cancer immunotherapy. However, it is unclear how the difference in TNFR costimulation contributes to T cell responses. In this study, to clarify the functional significance of four different TNFRs, OX40, 4-1BB, CD27 and GITR, we prepared corresponding single-chain TNF ligand proteins (scTNFLs) connected to IgG Fc domain with beneficial characteristics, i.e., Fc−scOX40L, Fc−sc4-1BBL, Fc−scCD27L (CD70) and Fc−scGITRL. Without intentional cross-linking, these soluble Fc−scTNFL proteins bound to corresponding TNFRs induced NF-kB signaling and promoted proliferative and cytokine responses in CD4^+^ and CD8^+^ T cells with different dose-dependencies in vitro. Mice injected with one of the Fc−scTNFL proteins displayed significantly augmented delayed-type hypersensitivity responses, showing in vivo activity. The results demonstrate that each individual Fc−scTNFL protein provides a critical costimulatory signal and exhibits quantitatively distinct activity toward T cells. Our findings provide important insights into the TNFR costimulation that would be valuable for investigators conducting basic research in cancer immunology and also have implications for T cell-mediated immune regulation by designer TNFL proteins.

## 1. Introduction

T cell activation requires the first signal via the T cell receptor (TCR)/CD3 by the recognition of peptide-MHC complexes and the second signal, termed the costimulatory signal, via the costimulatory receptors by the interaction with their specific ligands. The members of the TNF receptor superfamily (TNFRSF), OX40 (TNFRSF4), 4-1BB (TNFRSF9), CD27 (TNFRSF7) and GITR (TNFRSF18), expressed by CD4^+^ and CD8^+^ T cells are key costimulatory receptors and play vital roles for the regulation of T cell responses [1,2,3,4,5]. OX40, 4-1BB, CD27 and GITR are activated by the interaction with their cognate ligands of the TNF superfamily (TNFSF), OX40L (TNFSF4), 4-1BBL (TNFSF9), CD27L (CD70, TNFSF7) and GITRL (TNFSF18), respectively, expressed by antigen-presenting cells (APCs) such as dendritic cells, B cells and macrophages. The TNFSF-TNFRSF interactions are temporally and spatially regulated in the course of immune responses and regulate not only the initial phases of T cell responses to augment proliferation, differentiation and survival but also the later phases of effector/memory T cell responses [1,6,7,8,9,10,11]. Furthermore, signals mediated by the TNFRSF molecules play critical roles in immune mediated diseases [1,3,9,10,11,12,13,14]. Thus, the TNFSF-TNFRSF interactions have been recognized as therapeutic targets for intervention in immune diseases.

Our understanding of costimulatory TNFSF-TNFRSF interactions has progressed, and the wealth of experimental results are currently applied to the development of biologics or gene engineering technologies [15,16,17,18,19,20,21]. Despite rational strategies to generate biotherapeutics targeting the TNFSF-TNFRSF molecules, these efforts are frequently hindered by a lower performance in clinical trials [22,23,24]. We think that more fundamental analyses are still required to understand the role of costimulatory TNFSF-TNFRSF interactions in T cell responses. Unlike antagonists, where the function of molecules is basically to block the receptor-ligand interaction, the action of agonists is more diverse and their regulatory mechanisms are difficult to generalize. Thus, it is important to attain basic information on how costimulatory TNFRSF molecules exhibit their activities through interaction with agonists and what is the relative contribution of each TNFRSF molecule to others in regulating T cell responses. We believe that deciphering these questions would be important for the future development of novel TNFR-targeted immunotherapies.

OX40L, 4-1BBL, CD70 and GITRL are structurally and evolutionarily interconnected [25,26,27], and a C-terminal TNF homology domain (THD) of these TNFSF molecules noncovalently assembles into active homotrimers on the surface of APCs that stimulate corresponding TNFRSFs on T cells [28]. Although these TNFSF molecules play dominant roles in the activation of T cells, there are limited studies to directly evaluate their relative efficacy on T cells. A soluble TNFSF molecule in a single-chain format, scTNFL, can be generated by genetically connecting three extracellular THDs with two polypeptide linkers [29,30,31,32,33,34,35]. By using this sophisticated methodology, we prepared scTNFL proteins with immunoglobulin G (IgG) Fc domain, i.e., Fc−scOX40L, Fc−sc4-1BBL, Fc−scCD70, and Fc−scGITRL, to clarify how these TNFSF molecules regulate T cell responses via their interaction with corresponding TNFRSF molecules.

In the present study, we showed that these four Fc−scTNFL proteins bound to CD4^+^ and CD8^+^ T cells, which led to the induction of cell proliferation and cytokine release with different dose dependencies. Fc−scGITRL induced a potent T cell recall response in vivo. The fundamental information obtained in this study provides important insights into TNFR costimulation that would be helpful for investigators conducting fundamental immunological research and has implications for T cell-mediated immune regulation by engineered TNF proteins.

## 2. Materials and Methods

### 2.1. Plasmid

A pREP4 vector (Invitrogen, Waltham, MA, USA) containing a cDNA encoding mouse *Ox40l* (*Tnfsf4*) was previously described [36]. A qCR^TM^-Blunt II-TOPO^TM^ vector (#450245, Thermo Fisher, Waltham, MA, USA) containing a cDNA encoding mouse *Cd70* or mouse *41bb* (*Tnfrsf9*) was previously described [37]. PCR primers were purchased from Fasmac Co., Ltd. (Kanagawa, Japan). Based on the cDNA sequences of mouse *4-1bbl* (*Tnfsf9*, MN_009404.3), mouse *Gitrl* (*Tnfsf18*, BC137814.1), mouse *Cd27* (NM_001033126.2), mouse *Gitr* (*Tnfrsf18*, NM_009400.3) and mouse dihydrofolate reductase (*Dhfr*) (L26316.1), each cDNA of the entire coding region was amplified with PCR using specific primers and PrimeSTAR^TM^ GXL DNA polymerase (R050A, Takara Bio, Shiga, Japan) or KOD FX Neo^TM^ DNA polymerase (KFX-201, TOYOBO, Osaka, Japan), and the resulting PCR fragment was inserted into the vector qCR^TM^-Blunt II-TOPO^TM^. cDNA of the entire coding region of *Gitr* or *Dhfr* was ligated into a pcDNA3.1/V5-His A vector (Invitrogen). For construction of a gene for scOX40L, three copies of a gene corresponding to an extracellular THD region of *Ox40l* (^51^Ser−^198^Leu) were connected to each other with two oligonucleotides corresponding to a peptide linker GGGSGGG ([ggc gga ggc tca ggc ggt gga] and [gga gga gga tcc gga ggt gga]). The entire scOX40L gene with 5′-KpnI and 3′-AgeI restriction enzyme sites, (KpnI)−Ox40l−GGGSGGG−Ox40l−GGGSGGG−Ox40l−(AgeI), was synthesized by gBlocks^TM^ (Integrated DNA Technologies, Coralville, IA, USA). PCR DNA fragments corresponding to a PA-peptide (GVAMPGAEDDVV) tag (ggc gtt gcc atg cca ggt gcc gaa gat gat gtg gtg) and a His_6_-peptide (HHHHHH) tag (cat cat cac cat cac cat) were attached to both ends of the scOX40L gene by recombinant PCR (Appendix A). By using In-Fusion cloning (638947, In-Fusion Snap Assembly Master Mix, Takara Bio), the PA−scOX40L−His_6_ gene was inserted into the C-terminus of the Fc region (hinge, CH2 and CH3 domains) of human IgG1 genomic DNA (Appendix A) in the pEF−Fc expression vector [38], which was derived from the pEF-BOS vector [39]. The entire Fc−PA−scOX40L−His_6_ gene was amplified with PCR using primers that added a 5′-EcoRI site in forward primer and a 3′-BglII site in reverse primer, and the resulting PCR fragment was digested with EcoRI (R3101, New England Biolabs, Ipswich, MA, USA) and BglII (R0144, New England Biolabs), followed by ligation into the mammalian expression vector pCAGGS (GenBank accession number LT727518.1) [40].

Genes for sc4-1BBL, scCD70 and scGITRL were constructed as follows. Three PCR fragments containing an identical nucleotide sequence corresponding to an extracellular THD region of *4-1bb* (^122^Arg−^309^Glu)*, Cd70* (^45^Ser−^195^Pro) or *Gitr* (^42^Thr−^173^Ser) were amplified using three different primer pair sets. The first PCR fragment was amplified with a forward primer containing a 5′-KpnI site and a reverse primer containing a 3′-BamHI site within the GGGSGGG linker. The second PCR fragment was amplified with a forward primer containing a 5′-BamHI site within the GGGSGGG linker (ggt gga gga tcc ggt gga ggt) and a reverse primer containing a 3′-BspEI site within the GGGSGGG linker. The third PCR fragment was amplified with a forward primer containing a 5′- BspEI site within the GGGSGGG linker (gga ggc ggt tcc gga ggt ggg) and a reverse primer containing a 3′-AgeI site. The gel purified three PCR fragments which were digested with BamHI (R3136, New England Biolabs) and BspEI (R0540, New England Biolabs) were connected to each other using Ligation High^TM^ (LGK-101, TOYOBO, Osaka, Japan). After amplification with PCR, the entire gene for sc4-1BBL, scCD70 or scGITRL, (KpnI)−THD−GGGSGGG−THD−GGGSGGG−THD−(AgeI), was ligated into qCR^TM^-Blunt II-TOPO^TM^. The pCAGGS vector encoding Fc−PA−sc4-1BBL−His_6_ was constructed by replacing the gene for scOX40L in the pCAGGS vector encoding Fc−PA−scOX40L−His_6_ with the gene for sc4-1BBL in qCR^TM^-Blunt II-TOPO^TM^ vector, after enzymatic digestion with KpnI (R3142, New England Biolabs) and AgeI (R3552, New England Biolabs) (Appendix A). The pCAGGS vector encoding Fc−PA−scCD70−His_6_ or Fc−PA−scGITRL−His_6_ was constructed in the same manner described above (Appendix A).

A plasmid encoding mouse OX40 extracellular region (^21^Thr-^211^Pro) and human IgG1-Fc (mouse OX40-Fc) in the pEF-Fc vector [41], and a plasmid encoding mouse 4-1BB extracellular region (^24^Val−^187^Leu) and human IgG1-Fc (mouse 4-1BB-Fc) in the pEF-Fc vector [37] were previously described. A plasmid encoding mouse CD27 extracellular region (^24^Pro−^182^Arg) and human IgG1-Fc (mouse CD27-Fc) in the pEF-Fc vector was constructed in this study, and a PCR fragment (gac tac aag gat gac gat gac aag ctc gat gga gga tac cca tac gat gtt cca gat tac gct) corresponding to a Flag−HA tag peptide (DYKDDDDKLDGGYPYDVPDYA) was inserted into the 3′ end of the CD27−Fc gene by In-Fusion cloning.

### 2.2. Recombinant Protein

For production of recombinant proteins, the expression vector was transfected into HEK293T cells (CRL-3216, ATCC, Manassas, VA, USA) using polyethyleneimine (PEI) (408727, Merck, Burlington, MA, USA), and cells were cultured in DMEM (043-30085, FUJIFILM Wako Pure Chemical Cooperation, Osaka, Japan) supplemented with 2% fetal calf serum (FCS), 100 U/mL penicillin and 100 µg/mL streptomycin for 5 to 7 days [41,42]. Alternatively, the expression vector was transfected into DG44-CHO cells (Dhfr negative, CRL-9096, ATCC), and a stable DG44-CHO cell clone was established. Briefly, DG44-CHO cells were cultured in IMDM (098-06465, FUJIFILM) supplemented with 0.1 mM hypoxanthine, 0.016 mM thymidine, 2 µM methotrexate (MTX), 10% FCS, 100 U/mL penicillin and 100 µg/mL streptomycin, and cells were transfected with the pcDNA3.1 vector encoding *Dhfr* plus neomycin resistance genes and the pCAGGS vector encoding Fc−scOX40L, Fc−sc4-1BBL, Fc−scCD70 or Fc−scGITRL at 1:10 ratio. The transfected DG44-CHO cells were cultured in complete IMDM supplemented with 0.5 mg/mL G418 without hypoxanthine/thymidine/MTX. To induce a high degree of gene amplification, MTX concentration was increased in a stepwise manner from 50 to 800 nM, and single-cell clones were isolated from heterogenous cell pools by limiting dilution. To select cell clones producing higher levels of recombinant proteins, culture supernatants were screened by sandwich enzyme-linked immunosorbent assay (ELISA). Selected DG44-CHO cell clones were cultured either in IMDM containing 2% FCS or BalanCD^TM^ CHO GROWTH A/BalanCD^TM^ CHO FEED4 (554-34287, 530-94421, FUJIFILM).

The nonserum medium containing recombinant Fc−scTNFL protein was applied to a HiTrap^TM^ rProtein A FF column (17507901, Cytiva, Tokyo, Japan). After rinsing the column with phosphate-buffered saline (PBS), the bound Fc protein was eluted with elution buffer (80 mM glycine-HCl, 0.8 M arginine, pH 3.0), and protein fractions were immediately neutralized with 1 M Tris-HCl (pH 9.0). The elute was dialyzed against PBS, concentrated with Amicon Ultra-4 (UFC801024, Merck) and filtered with Millex-GP (0.22 µm, SLGPR33RS, Merck). Total protein concentration was determined by bicinchoninic acid (BCA) assay (297-73101, FUJIFILM). 

Protein samples were reduced with 4× Laemmli sample buffer (240 mM Tris-HCl, pH 6.8, 40% glycerol, 8% sodium dodecyl sulfate (SDS), 0.1% bromophenol blue, and 4% 2-mercaptoethanol (2-ME)) at 100 °C for 1 to 10 min and were analyzed by SDS-polyacrylamide gel electrophoresis (PAGE), followed by Coomassie blue staining (EzStain Aqua, AE-1340, ATTO Corp., Tokyo, Japan). 

Fc−scTNFL proteins were produced with average yield of 10 to 100 mg per liter in the culture supernatant of stably transfected CHO cells.

### 2.3. Flow Cytometry

A DG44-CHO cell line stably expressing mouse Gitr was established by transducing the pcDNA3.1/V5-His A vector encoding *Gitr*, and the expression of Gitr was confirmed by staining the cell line with 0.25 µg/mL of biotin-anti-GITR antibody (DTA-1, 126305, BioLegend, San Diego, CA, USA) and 0.25 µg/mL of fluorescein isothiocyanate (FITC)-streptavidin (405201, BioLegend) in FACS buffer (0.2% bovine serum albumin (BSA) and 0.02% sodium azide in PBS). For detection of GITRL−GITR interaction, the Gitr^+^ CHO cell line was firstly incubated with 5 µg/mL of Fc−scGITRL, secondarily with 0.07 µg/mL of biotin-anti-human IgG (109-065-190, JacksonImmunoResearch, West Grove, PA, USA), and thirdly with 0.25 µg/mL of FITC-streptavidin. 

The following antibodies were used for staining of OX40, 4-1BB, CD27 and GITR expressed by CD4^+^ and CD8^+^ T cells purified form C57BL/6 mouse spleens: 1 µg/mL of allophycocyanin (APC)-anti-OX40 (OX86, 119413, BioLegend), 1.25 µg/mL of biotin-anti-4-1BB (17B5, 106103, BioLegend), 1.25 µg/mL of biotin-anti-CD27 (LG.3A10, 124205, BioLegend), 1.25 µg/mL of biotin-anti-GITR (DTA-1, 126305, BioLegend) and 0.25 µg/mL of phycoerythrin (PE)-streptavidin (405203, BioLegend). The binding of Fc−scTNFL proteins (5 µg/mL) to CD4^+^ and CD8^+^ T cells was visualized by 0.07 µg/mL of biotin-anti-human IgG and 0.25 µg/mL of PE-streptavidin. Data were acquired on a FACSCanto II (BD Bioscience, Franklin Lakes, NJ, USA) and analyzed with Flowing software 2.5.1 (http://flowingsoftware.btk.fi/, URL (accessed on 19 September 2021), Turku Bioscience, Turku, Finland).

### 2.4. ELISA

For screening of cell clones producing higher levels of Fc−scTNFL proteins, culture supernatants were added to ELISA plates (439454, Thermo Fisher, Waltham, MA, USA) precoated with 0.5 µg/mL of mouse anti-His tag IgG (011-23091, FUJIFILM), and the binding between anti-His tag IgG and Fc−scTNFL was visualized by 0.07 µg/mL of biotin-anti-human IgG and HRP-streptavidin (405210, BioLegend) at a 1:3000 dilution.

For detection of OX40L−OX40 interaction, ELISA wells were coated with 0.5 µg/mL of OX40−Fc, and soluble Fc−scOX40L was added to the wells, followed by addition of 0.2 µg/mL of rat anti-PA tag IgG (NZ-1, 016-25863, FUJIFILM) and 0.07 µg/mL of anti-rat IgG−HRP (112-035-167, JacksonImmunoResearch). 4-1BBL−4-1BB interaction was evaluated in a similar way. For detection of CD70−CD27 interaction, ELISA wells were coated with 10 µg/mL of Fc−scCD70, and soluble CD27−Fc was added to the well, followed by addition of 0.1 µg/mL of mouse anti-Flag tag antibody (DYKDDDDK, 1E6, 018-22381, FUJIFILM) and anti-mouse IgG−HRP (330, MBL Co. LTD., Nagoya, Japan) at a 1:5000 dilution.

The following reagents were used for measurement of cytokines in T cell culture supernatants: 0.5 µg/mL of anti-IL-2 (JES6-1A12, 503701, BioLegend) capture antibody, 0.25 µg/mL of biotin-anti-IL-2 (JES6-5H4, 503803, BioLegend) detection antibody, 0.5 µg/mL of anti-IFN-g (R4-6A2, 505702, BioLegend) capture antibody, 0.25 µg/mL of biotin-anti-IFN-g (XMG1.2, 505804, BioLegend) detection antibody, HRP streptavidin (405210, BioLegend) at a 1:3000 dilution, and 3,3′,5,5′-tetramethyl benzidine (TMB, 421101, BioLegend). The absorbance was measured at 450 nm using FilterMax F5 (Molecular Devises, San Jose, CA, USA). 

### 2.5. Immunoblot Analysis

An OX40^+^ T cell hybridoma cell was previously reported [36,43,44]. Cells were lysed for 30 min in ice-cold radioimmunoprecipitation assay (RIPA) buffer (20 mM Tris-HCl, pH 7.5, 150 mM NaCl, 2 mM EDTA, 1% NP-40, 50 mM NaF, 1 mM Na_3_VO_4_, 1% sodium deoxycholate, and 0.1% sodium dodecyl sulfate) with a protease inhibitor mixture (P8340, Merck). The lysate was mixed with 4× Laemmli sample buffer and boiled for 5 min at 100 °C. Samples were separated by SDS-PAGE transferred onto polyvinylidene difluoride membranes (034-25663, FUJIFILM) and analyzed by immunoblot with anti-IkBa (L35A5, 4814, Cell Signaling Technology, Danvers, MA, USA) at a 1:2000 dilution and anti-b-actin (C4, MAB1501, Merck) at a 1:2000 dilution. The reaction was visualized with a chemiluminescence detection system using Immobilon Classico Western HRP substrate (WBLUC0500, Merck) and LAS-4000mini (FUJIFILM).

### 2.6. CD4^+^ and CD8^+^ T Cell Stimulation Assay

C57BL/6 mice (Japan SLC, Shizuoka, Japan) were bred under specific pathogen-free conditions. Animal experimental protocols were approved by the Animal Care and Use Committee of the University of Toyama (Approval Number: A2021PHA-1 and A2021PHA-11) and conducted in accordance with the Institutional Animal Experiment Handling Rules of the University of Toyama. Splenic CD4^+^ and CD8^+^ T cells from both male and female mice aged between 10 and 17 weeks were separated with CD4 (L3T4) microbeads (130-117-043, Miltenyi Biotec, Bergisch Gladbach, Germany) and CD8a (Ly-2) microbeads (130-117-044, Miltenyi Biotec), respectively. T cells were cultured in RPMI1640 (189-02025, FUJIFILM) supplemented with 10% FCS, 100 U/mL penicillin, and 100 µg/mL streptomycin, 50 µM 2-ME. T cells (5 × 10^4^ cells/well) were plated in 96-well F-bottomed culture plates precoated with anti-CD3e antibody (low-endotoxin, azide free, 145-2C11, 100340, BioLegend) in PBS at concentrations of 0.3, 1, 3, 10 µg/mL in the presence or absence of soluble Fc−scTNFL proteins at concentrations of 0.3, 1, 3, 10 µg/mL. Cell proliferation was assessed with a 3-(4,5-dimethylthiazol-2-yl)-2,5-diphenyl-tetrazolium (MTT) (M2128, Merck) assay. The water-insoluble MTT formazan was solubilized dimethyl sulfoxide (DMSO), and the absorbance was measured at 540 nm using FilterMax F5 (Molecular Devises). 

For evaluation of the expression of TNFRSF molecules on activated T cells in flow cytometry, purified CD4^+^ or CD8^+^ T cells were stimulated with 1 µg/mL of anti-CD3 antibody, 1 µg/mL of anti-CD28 antibody (low-endotoxin, azide free, 37.51, 102116, BioLegend) and 1 ng/mL of IL-2 (200-02, PeproTech, Rocky Hill, NJ, USA) for 3 days.

### 2.7. Delayed-Type Hypersensitivity Response

Delayed-type hypersensitivity (DTH) response was evaluated as previously described [37,42]. C57BL/6 mice from both male and female mice aged between 12 and 14 weeks were immunized subcutaneously at the tail with 200 µL of 1.25 mg/mL methyl BSA (mBSA) (A1009, Merck) emulsified with complete Freund’s adjuvant (CFA) (F5881, Merck) on day 0. Seven days after the immunization, mice were challenged subcutaneously in a footpad with 30 µL of 7 mg/mL mBSA in PBS plus 10 µL of PBS or 1000 µg/mL of Fc−scGITRL. An equal volume of PBS was injected into another footpad as a control. One day after the challenge, footpad thickness was measured with a digital caliper (Shinwa Rules Co., Ltd., Japan). The magnitude of the DTH response was determined as follows: [footpad swelling (%)] = ([footpad thickness of mBSA-injected footpad (mm)] − [footpad thickness of PBS-injected footpad (mm)]) ÷ [footpad thickness of PBS-injected footpad (mm)] × 100. Each frozen block of footpad was cut into 5-μm-thick sections. After fixing with 100% ethanol and 4% paraformaldehyde, samples were stained with hematoxylin and eosin (H&E). Images of each footpad section were taken using a fluorescence microscope (BZ-X800, Keyence, Osaka, Japan).

### 2.8. Statistical Analysis

Statistical significance was determined by a two-tailed unpaired Student’s *t* test for two groups, with the assumption of normal distribution of data and equal sample variance. A *p* value of less than 0.05 was considered statistically significant.

## 3. Results

### 3.1. Specific Binding of Fc−scTNFL Fusion Protein to Corresponding TNFR on T Cells

TNFRSF molecules on T cells and TNFSF molecules on APCs interact during the course of immune responses, and these physiologically relevant TNFRSF-TNFSF interactions regulate immunity and diseases. The activation of TNFRSF molecules plays important roles in the control of T cell immunity, and thus it is very important to understand what types of molecules can be a substitute for naturally expressing TNFSF molecules on APCs. The information would be beneficial for the design of novel therapeutic agents for infections and cancers.

To understand how the activity of TNFRSF molecules expressed by CD4^+^ and CD8^+^ T cells is controlled by engineered TNFSF molecules, we prepared four scTNFL proteins, scOX40L, sc4-1BBL, scCD70 and scGITRL, in which three identical THD protomers were covalently connected to each other with two GGGSGGG peptide linkers to support the organization of the trimer structure of TNF ligand protein. For the convenience of protein detection and purification, a PA-peptide tag [45] and a His_6_-peptide tag were attached to both ends of the scTNFL, and additionally, IgG Fc domain was connected to the N-terminus of the PA−scTNFL−His_6_ to create Fc−scTNFL fusion proteins, i.e., Fc−scOX40L, Fc−sc4-1BBL, Fc−scCD70 and Fc−scGITRL (Figure 1). These four Fc−scTNFL proteins purified from culture supernatants displayed a SS-linked dimer structure in SDS-PAGE (Figure 2). Thus, one Fc−scTNFL molecule has two TNFL trimer units and forms a hexameric structure.

Next, to evaluate the binding activity of Fc−scTNFL proteins to corresponding TNFRs, each soluble Fc−scTNFL was added to ELISA wells precoated with OX40-Fc or 4-1BB-Fc. Fc−scOX40L, but not Fc−sc4-1BBL, Fc−scCD70 or Fc−scGITRL bound to OX40−Fc (Figure 3A). Similarly, Fc−sc4-1BBL, but not Fc−scOX40L, Fc−scCD70 or Fc−scGITRL bound to 4-1BB−Fc (Figure 3B). Although we failed to detect binding activity of Fc−scCD70 and Fc−scGITRL to corresponding receptors in the same assay, plate-coated Fc−scCD70, but not Fc−scOX40L, Fc−sc4-1BBL or Fc−scGITRL, interacted with soluble CD27−Fc (Figure 3C). Additionally, soluble Fc−scGITRL, but not Fc−scOX40L, Fc−sc4-1BBL or Fc−scCD70 bound to cell surface GITR (Figure 3D). These results demonstrate that all four Fc−scTNFL proteins exhibit specific binding activity to their corresponding TNFRs and no cross-reactivity with irrelevant TNFRs.

It has been demonstrated that the expression of TNFRs on CD4^+^ and CD8^+^ T cells shows variation and is positively and negatively regulated by activating signals via the TCR and CD28. OX40 and 4-1BB are significantly induced on activated CD4^+^ and CD8^+^ T cells following T cell activation [46,47,48,49,50]. In contrast, CD27 is highly expressed by naïve or resting CD4^+^ and CD8^+^ T cells and can be downregulated following T cell activation [3,8,51]. GITR is present on naïve or resting CD4^+^ and CD8^+^ T cells and can be upregulated after T cell activation [52,53,54].

Based on the expression profile described above, we firstly confirmed the levels of OX40, 4-1BB, CD27 and GITR on CD4^+^ and CD8^+^ T cells using staining antibodies for these TNFRs before and after T cell activation. The expression of OX40 and 4-1BB on CD4^+^ and CD8^+^ T cells was upregulated after T cell activation (Figure 4A,C). In contrast, CD27 was highly expressed by unstimulated CD4^+^ and CD8^+^ T cells, and the expression was greatly downregulated after T cell activation (Figure 4A,C). GITR was also expressed by unstimulated CD4^+^ and CD8^+^ T cells, and the expression was maintained or upregulated after T cell activation (Figure 4A,C). 

Consistent with the above results, all four Fc−scTNFL proteins bound to the surface of CD4^+^ and CD8^+^ T cells (Figure 4B,D), and the staining profile of Fc−scTNFL proteins was comparable with that of staining antibodies (Figure 4A–D). 

Collectively, these results demonstrate that all four Fc−scTNFL proteins consistently exhibit specific binding activity toward corresponding TNFRs expressed by CD4^+^ and CD8^+^ T cells.

### 3.2. NF-kB Activation by Fc−scTNFL

The classical NF-kB pathway plays essential roles in T cell activation. The degradation of IkBa protein has been used as a representative marker for the activation of the classical NF-kB signaling. Costimulatory TNFRs including OX40, 4-1BB, CD27 and GITR are major activators of the NF-kB pathway [1,55,56]. Our group previously established a T cell hybridoma cell line expressing OX40 [36,43,44]. This cell also endogenously expressed CD27 and GITR at a steady state (Figure 5A), and 4-1BB could be induced after stimulation with PMA and ionomycin (Figure 5A,C). The binding profile of Fc−scTNFL proteins was comparable with that of antibodies for TNFRs (Figure 5A). To evaluate the NF-kB pathway activation, this T cell was cultured with respective Fc−scTNFL proteins.

Upon stimulation with soluble Fc−scOX40L, IkBa was degraded in a time-dependent manner in this T cell (Figure 5B). Similarly, stimulation of T cells with Fc−scCD70 or Fc−scGITRL resulted in IkBa degradation (Figure 5B), and Fc−sc4-1BBL induced the degradation of IkBa in activated T cells (Figure 5C,D). Thus, all Fc−scTNFL proteins activate the classical NF-kB pathway mediated by OX40, 4-1BB, CD27 and GITR.

### 3.3. Activation of Costimulatory Signaling Pathway by Fc−scTNFL in CD4^+^ and CD8^+^ T Cells

Signaling through costimulatory receptors augments the TCR/CD3 signals that ensure T cell proliferation and cytokine production. The costimulatory signals derived from TNFSF-TNFRSF interactions, OX40L−OX40, 4-1BBL−4-1BB, CD70−CD27 and GITRL−GITR, play critical roles in the activation of CD4^+^ and CD8^+^ T cells [1,3,4,5]. 

To evaluate costimulatory activity of Fc−scTNFL proteins, CD4^+^ or CD8^+^ T cells purified from mouse spleens were stimulated with a fixed concentration (10 µg/mL) of plate-bound anti-CD3 agonistic antibody and increasing concentrations of soluble Fc−scTNFL proteins. Upon stimulation of CD4^+^ T cells with anti-CD3, additional Fc−scOX40L significantly promoted IL-2 and IFN-g production and cell proliferation (Figure 6A). Similarly, Fc−sc4-1BBL, Fc−scCD70 or Fc−scGITRL dose-dependently induced cytokine release and proliferation of CD4^+^ T cells (Figure 6A), showing a potent costimulatory activity of Fc−scTNFL proteins for CD4^+^ T cells. Notably, Fc−scGITRL could efficiently increase IL-2 from CD4^+^ T cells even at 0.3 µg/mL (Figure 6A). As with CD4^+^ T cells, Fc−sc4-1BBL, Fc−scCD70 and Fc−scGITRL also significantly augmented cytokine and proliferative responses of CD8^+^ T cells (Figure 6B). In contrast, Fc−scOX40L displayed a less potent activity for CD8^+^ T cells, as compared to CD4^+^ T cells. Interestingly, Fc−sc4-1BBL could efficiently promote the production of IFN-g and proliferation, as compared to other Fc−scTNFL proteins (Figure 6B). Thus, these results demonstrate that Fc−scTNFL proteins may have redundant and specific functions in promoting costimulatory signaling pathways in CD4^+^ and CD8^+^ T cells.

To understand how the costimulatory signal mediated by Fc−scTNFL proteins is dependent on the TCR/CD3 signal, CD4^+^ or CD8^+^ T cells were stimulated with increasing concentrations of anti-CD3 in the presence or absence of a fixed concentration (1 µg/mL) of respective Fc−scTNFL proteins. This analysis revealed that higher concentrations of anti-CD3 were required for Fc−scTNFL proteins to increase cytokine production and cell proliferation in both CD4^+^ and CD8^+^ T cells (Figure 7A,B). Notably, Fc−scGITRL exhibited better costimulatory activity against CD4^+^ T cells as compared to other Fc−scTNFL proteins (Figure 7A). Fc−scGITRL induced the highest amount of IL-2 from CD4^+^ T cells (Figure 7A), whereas Fc−sc4-1BBL produced the largest amount of IFN-g from CD8^+^ T cells (Figure 7B).

These results show that all four Fc−scTNFL proteins retain an active TNF ligand structure to be able to agonize corresponding TNFRs expressed by CD4^+^ and CD8^+^ T cells and efficiently promote the production of cytokines and proliferation in a dose-dependent manner.

### 3.4. DTH Response Mediated by Fc−scGITRL

To demonstrate costimulatory activity of Fc−scTNFL proteins in vivo, mice were immunized with mBSA on day 0 and challenged with mBSA with or without Fc−scTNFL on day 7 to induce DTH response, classified as type IV hypersensitivity associated with CD4^+^ T cell inflammation, as determined by footpad swelling on day 8. In our preliminary experiments, injection of one of the Fc−scTNFL proteins, Fc−scGITRL, in the challenge phase showed a trend toward increased DTH response, as compared to other Fc−scTNFL proteins. We thought that this might be related to the higher costimulatory activity of Fc−scGITRL for CD4^+^ T cells as shown in Figure 6A and Figure 7A. For this reason, we decided to carefully examine whether Fc−scGITRL could impact the DTH response.

The footpad swelling was significantly enhanced in mice injected with mBSA plus Fc−scGITRL (G2) compared to mBSA alone (G1) within 1 day after the challenge (Figure 8A,B). Fc−scGITRL induced the increment in the epidermal thickness of footpad skin (Figure 8C and Appendix A). Spleen cells from mice challenged with mBSA and Fc−scGITRL (G2) showed significantly higher proliferative responses to mBSA as compared to those with mBSA alone (G1) (Figure 8D). These results indicate that Fc−scGITRL rapidly augments T cell-dependent inflammation in the challenge phase of DTH response.

Taken together, the results obtained in this study demonstrate that Fc−scTNFL proteins efficiently enhance CD4^+^ and CD8^+^ T cell activation with different dose-dependencies and suggest that a rational arrangement of TNFR agonism would fulfill roles in promoting the activation, differentiation and survival of antigen-primed T cells.

## 4. Discussion

Over the past few decades, the TNFSF-TNFRSF of costimulatory molecules have been investigated to understand the important attributes of the T cell response, and especially the cognate interactions of OX40L-OX40, 4-1BBL-4-1BB, CD70-CD27 and GITRL-GITR have been shown to be critical for immune regulation by T cells. The aim of this study is to clarify the functional significance of four different TNFRSF molecules, OX40, 4-1BB, CD27 and GITR, in the regulation of the TNFR costimulation by using recombinant Fc−scTNFL proteins, i.e., Fc−scOX40L, Fc−sc4-1BBL, Fc−scCD70 and Fc−scGITRL. Without intentional cross-linking, soluble Fc−scTNFL proteins concurrent with anti-CD3 induced significant cytokine production and cell proliferation, as compared with anti-CD3 alone, showing that Fc-scTNFL proteins provide a potent costimulatory signal to T cells. The results demonstrate differential costimulatory activity of the scTNFL moiety of the fusion molecule and suggest that each individual cognate interaction quantitatively contributes to the TNFR costimulation that is essential for T cell activation. Our findings provide novel insights into the TNFSF-TNFRSF of costimulatory molecules in T cell responses and have important implications for the control of immune mediated diseases, including infections and cancers.

It has been a matter of concern how OX40, 4-1BB, CD27 and GITR regulate T cell responses. Expression levels of these TNFRs and their ligands are differentially controlled by the activation status of T cells and APCs, and these TNFRs might use different combinations of TNFR-associated factor (TRAF) family molecules to activate downstream signaling pathways. Thus, these costimulatory TNFRs may work cooperatively and/or independently. However, the functional significance of each individual TNFR for the activation of T cells has not been fully understood.

To clarify costimulatory activity of OX40, 4-1BB, CD27 and GITR, we decided to prepare corresponding Fc-scTNFL proteins, Fc−scOX40L, Fc−sc4-1BBL, Fc−scCD70 and Fc−scGITRL, respectively. For construction of the Fc−scTNFL gene, three copies of the extracellular THD genes were connected with two flexible glycine-serine-linker (GGGSGGG) genes, followed by linking to the C-terminus of the gene for the Fc fragment of human IgG1. The concatenated gene of Fc-scTNFL was inserted into the expression plasmid pCAGGS. The genetically engineered Fc-scTNFL proteins were purified from culture supernatants of CHO cells, followed by affinity chromatography purification with Protein A column. Accurate Fc dimer assembly of the Fc-scTNFL protein was confirmed by SDS-PAGE, indicating formation of a hexameric TNFL structure. All Fc-scTNFL proteins specifically interacted with their corresponding TNFRSF molecules and activated the NF-kB pathway, indicating the formation of an active TNF trimer structure. Finally, we verified that all four Fc-scTNFL proteins agonize the TNFRs expressed by CD4^+^ and CD8^+^ T cells. Our preliminary results suggest that the hexameric Fc-scTNFL molecule with two trimeric TNFL subdomains displayed more potent agonistic activity than one trimeric TNFL molecule (unpublished observation). Thus, the molecular format of scTNFL and IgG Fc domain would be generally beneficial for creating an agonist for TNFRSF molecules.

The expression levels of TNFRs on the surface of T cells are primarily important for the costimulatory function of TNFRs. It has been reported that OX40 expression on CD4^+^ and CD8^+^ T cells is enhanced by increasing engagement of the TCR/CD3 and that under physiological conditions, other costimulatory signals, including CD28, augment the expression of OX40. IL-1, IL-2, IL-4 and TNF-a also contribute to the prolonged expression of OX40. In this study, for the activation of CD4^+^ and CD8^+^ T cells in vitro we did not add anti-CD28 agonistic antibody in the culture (we added anti-CD28 antibody in the stimulation culture to optimally induce the upregulation of OX40 in the experiment of Figure 4). The lower concentration of anti-CD3 might not support the expression of OX40 in this in vitro culture setting. This may be the reason why we could only see increased cytokine and proliferative responses mediated by Fc-scOX40L at 10 µg/mL of anti-CD3 condition in Figure 7. A critical question would be how the expression level of TNFRSF molecules on T cells correlates with the activity of T cells. In the physiological setting, this might be influenced by the availability of antigens and cognate TNF ligands on APCs and by downregulation or shedding of TNF receptors in the case of CD27 [57,58]. It will be important to elucidate how the expression levels of costimulatory TNFRs have an impact in the T cell responses induced by Fc-TNFL proteins in more physiologically relevant conditions.

In this study, we could confirm a critical attribute of the 4-1BB costimulation for the activation of CD8^+^ T cells in terms of proliferation and cytokine production. Using cell lines expressing anti-CD3 antibody and membrane-bound TNFSF molecules, Kober et al. [59] reported that 4-1BBL showed the most potent activity for CD8^+^ T cells, as compared to OX40L, CD70, GITRL, CD30L or LIGHT. Using plate-immobilized HLA-peptide complex and TNFSF molecules, Neuyen et al. [57] examined the role of the TNFR costimulation in CD8^+^ T cells and demonstrated that the 4-1BB costimulation showed the strongest amplification of cytokine production, as compared to CD27, GITR or OX40. Warwas et al. [60] tested costimulatory activities of TNFSF molecules using bifunctional recombinant fusion proteins and demonstrated that 4-1BBL exhibited a better costimulatory capacity than OX40L or CD70. These studies show that the 4-1BBL−41BB interaction plays a more predominant role for the activation of CD8^+^ T cells.

Additionally, we characterized the functional significance of the GITR costimulation for the activation T cells. Although Kober et al. [59] reported that costimulatory function of cell-associated GITRL was significantly lower than 4-1BBL, CD70 or OX40L, engineered GITRL fusion proteins efficiently promote T cell responses [61,62,63,64]. We found that Fc−scGITRL stimulated a greater production of IL-2 from CD4^+^ T cells. Administration of Fc−scGITRL concurrently with antigen into mice in the challenge phase significantly promoted the DTH response as determined by footpad swelling, implying that the GITR costimulation critically augments recall T cell responses. Thus, it will be important to elucidate how respective Fc−scTNFL proteins control different types of T cell responses in vivo in future studies.

Introduction of costimulatory intracellular domains from TNFRSF molecules, including OX40, 4-1BB, CD27 and GITR, into chimeric antigen receptor (CAR) constructs for CAR-T cell therapy has been demonstrated to effectively improve the functionality of CAR-T cells, i.e., T cell longevity and antitumor effects [65]. Among the four TNFRSF molecules focused on in this study, the intracellular costimulatory domain of 4-1BB has been the most extensively studied. Second-generation CARs incorporating either 4-1BB or CD28 costimulatory domain have proven to be effective and have been widely used in the clinic. The 4-1BB domain is advantageous over the CD28 domain in terms of inducing beneficial activity of CAR-T cells, which might be explained by the enhanced mitochondrial biogenesis and oxidative metabolism [66]. Although characterization of CARs with costimulatory domains of OX40, CD27 and GITR is lagging behind in comparison to CARs with 4-1BB, these TNFRSF domains have been demonstrated to be potent CAR-T cell drivers [67,68,69,70,71]. Further research regarding rational selection and engineering of intracellular TNFRSF domains would be important to optimize CAR-T activity that leads to therapeutic benefit.

The interplay among the TNFSF-TNFRSF of costimulatory molecules has been suggested to play a vital role for immune regulation, and a rational arrangement of costimulatory signals via OX40, 4-1BB, CD27 and GITR might be important for establishing protective T cell-immunity for virus infections. Choi et al. [72] reported that B cells expressing Epstein-Barr virus (EBV) signaling molecule LMP1 had higher levels of CD70, OX40L and 4-1BBL on the cell surface and that the costimulatory signal mediated by these TNFSF molecules induced potent cytotoxic CD4^+^ and CD8^+^ T cell responses. This study suggests that CD70, OX40L and 4-1BBL on antigen-presenting B cells promote the differentiation and expansion of antigen-specific cytotoxic T cells that eradicate EBV-related cancers. In accordance with this, immune responses against EBV infection are compromised in patients with inborn errors of CD70, CD27, TNFSF9 (4-1BBL) or TNFRSF9 (4-1BB) [73,74,75,76,77,78]. The CD27 costimulation in CD8^+^ T cells initially promoted the expression of 4-1BB and subsequent activation of the 4-1BB costiumlation [57]. Concurrent activation of OX40 and 4-1BB pathways was effective for inducing cytotoxic CD4^+^ Th1 cells with tumoricidal function [79]. These studies suggest that an amalgam of signals via TNFRs augments and prolongs the duration of protective T cell responses. If this is correct, combination treatment of TNF ligand proteins may be advantageous for control of EBV infection and related cancers.

In summary, we demonstrated that soluble Fc−scTNFL proteins efficiently activated CD4^+^ and CD8^+^ T cells with different dose-dependencies. The results have implications for immune regulation by rationally engineered TNF ligand molecules. Cell surface presentation of scTNFL protein moieties might be a better way to control T cell-mediated immunity. A scTNFL domain as the antigen-binding domain of CAR was used for cancer immunotherapy [80]. If selected sets of scTNFLs are efficiently presented on the surface of APCs, these cells would be beneficial for the induction of antigen-specific immune responses. Fc-scTNFL molecules using in this study can be humanized, and the human scTNFL moiety presented on the surface of T cells or APCs might promote immune-mediated tumor destruction in a human setting. Further research will provide additional information regarding TNFR-targeted T cell regulation that may be advantageous for therapeutic intervention for infections and cancers.

## Figures and Tables

**Figure 1 cells-12-01596-f001:**
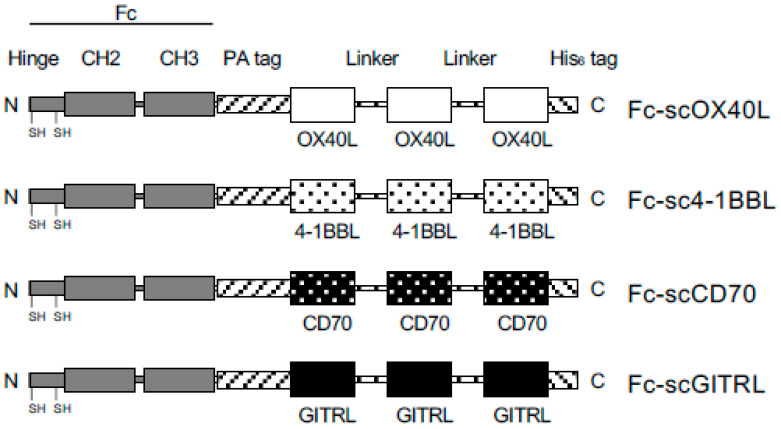
Schematic representation of Fc−scTNFL proteins. The domain structure of Fc [Hinge containing 2 cysteines, CH2 and CH3 domains], PA peptide tag (GVAMPGAEDDVV), extracellular TNF homology domain (THD) of OX40L (^51^Ser−^198^Leu) or 4-1BBL (^122^Arg−^309^Glu) or CD70 (^45^Ser−^195^Pro) or GITRL (^42^Thr−^173^Ser), GGGSGGG peptide linker, extracellular THD of OX40L or 4-1BBL or CD70 or GITRL, GGGSGGG peptide linker, extracellular THD of OX40L or 4-1BBL or CD70 or GITRL and His_6_ peptide tag (HHHHHH). Nucleotide sequences of Fc−scTNFL proteins are shown in Appendix A.

**Figure 2 cells-12-01596-f002:**
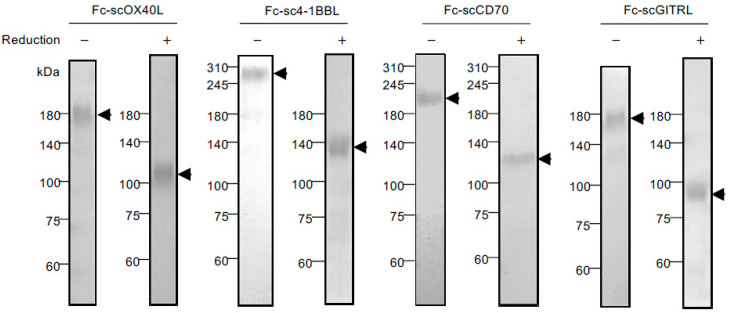
Electrophoretic analysis of Fc−scTNFL proteins. Coomassie-stained sodium dodecyl sulfate-polyacrylamide gel electrophoresis (SDS-PAGE) of Fc−scOX40L, Fc−sc4-1BBL, Fc−scCD70 and Fc−scGITRL under non-reducing (−) and reducing (+) conditions. The migration positions of molecular mass (kDa) markers are shown to the left of each gel. Arrow heads indicate positions of Fc-scTNFL proteins. The expected molecular weight of reduced Fc−scTNFL protein based on the amino acid sequence: Fc−scOX40L (79 kDa); Fc−sc4-1BBL (92 kDa); Fc−scCD70 (79 kDa); Fc−scGITRL (74 kDa). Data are from one experiment representative of at least two independent experiments with similar results.

**Figure 3 cells-12-01596-f003:**
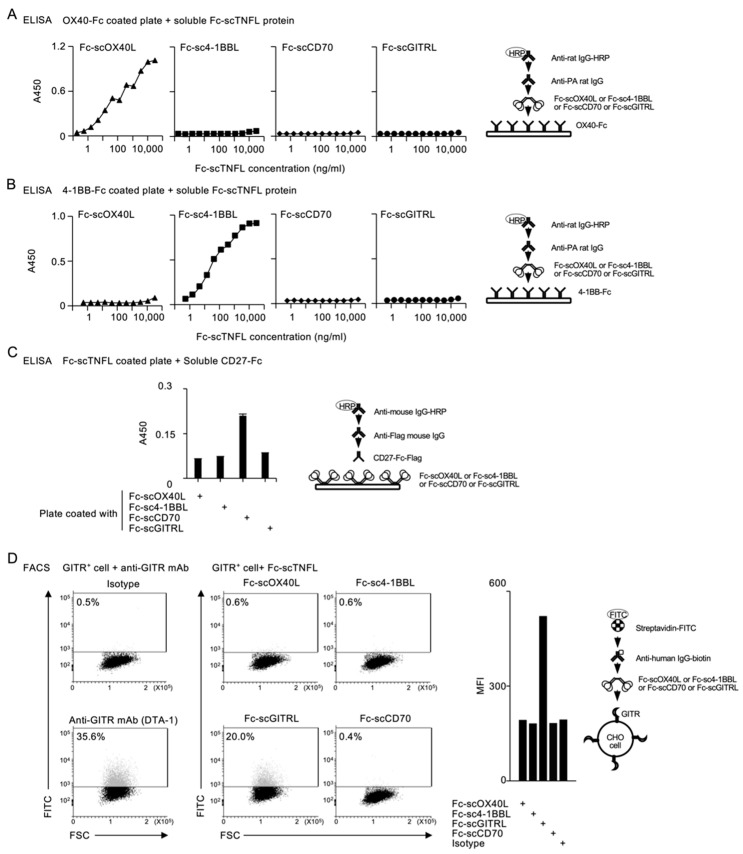
Binding specificity of Fc−scTNFL proteins to their corresponding receptors. (**A**–**C**) Validation of specific interactions between Fc−scOX40L and OX40−Fc (**A**), Fc−sc4-1BBL and 4-1BB−Fc (**B**) and Fc−scCD70 and CD27−Fc (**C**), evaluated by ELISA (average and standard deviation of triplicate wells in (**C**)). (**D**) Validation of specific interaction between Fc−scGITRL and cell surface GITR, evaluated by flow cytometry (middle) and expressed as mean fluorescent intensity (MFI) (right). GITR expression on CHO cells was confirmed by staining with anti-GITR antibody (DTA-1) (left). Numbers adjacent to outline areas indicate percent GITR^+^ cells. Data are from one experiment representative of at least two independent experiments with similar results.

**Figure 4 cells-12-01596-f004:**
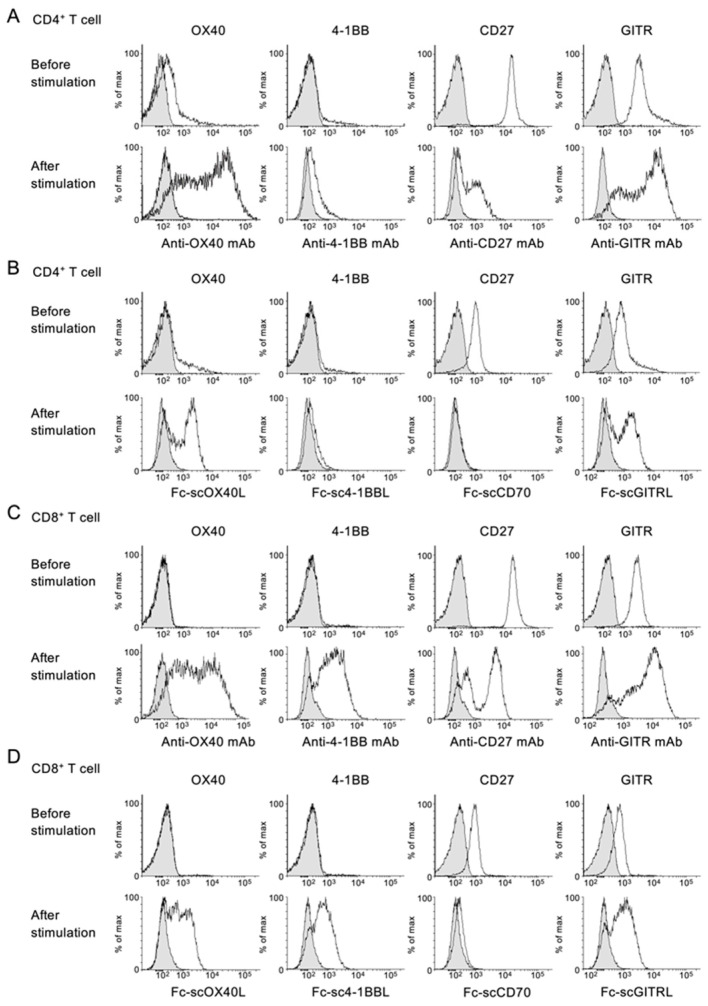
Binding activity of Fc−scTNFL proteins to CD4^+^ and CD8^+^ T cells. Splenic total CD4^+^ (**A**,**B**) and CD8^+^ (**C**,**D**) T cells were stained with indicated antibodies for TNFRs (**A**,**C**) or with indicated Fc−scTNFL proteins (**B**,**D**), evaluated by flow cytometry. CD4^+^ or CD8^+^ T cells were stained right after purification (before stimulation) or 3 days after stimulation with anti-CD3 and -CD28 antibodies plus IL-2 (after stimulation). Positive staining is shown by white histogram. Negative staining control is shown by shaded histogram. Data are from one experiment representative of at least two independent experiments with similar results.

**Figure 5 cells-12-01596-f005:**
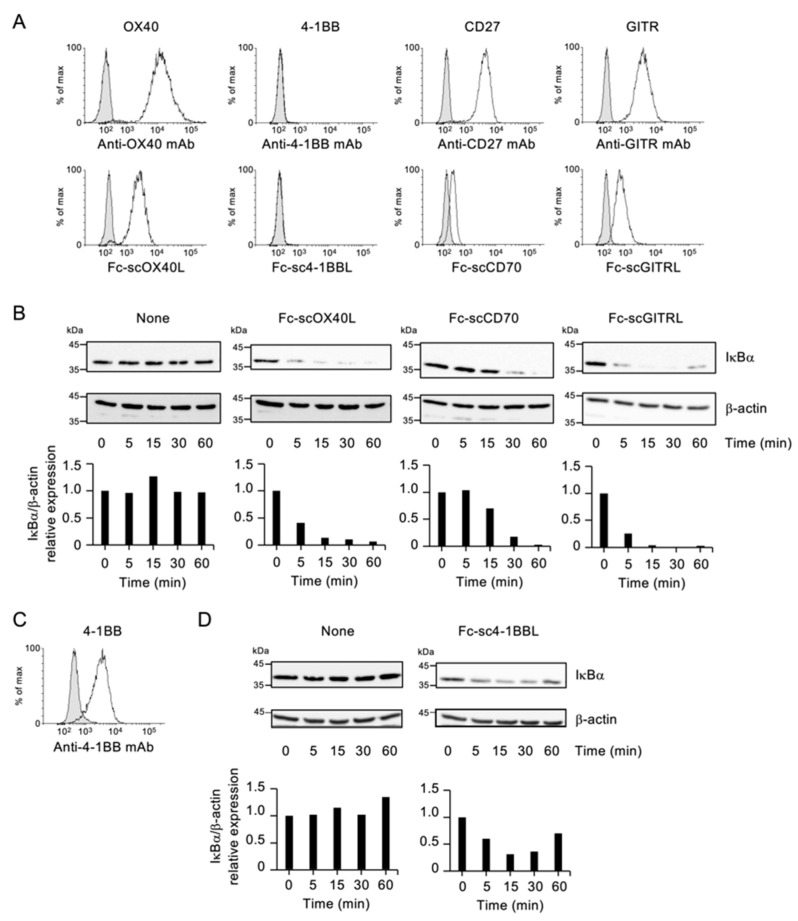
The NF-kB activation by Fc−scTNFL proteins. (**A**) Fc−scTNFL protein binding to cell surface TNFRs evaluated by flow cytometry. The expression of TNFRs on the surface of T cell hybridoma cells was evaluated using monoclonal antibodies to OX40, 4-1BB, CD27 or GITR (upper) or Fc−scTNFL proteins, Fc−scOX40L or Fc−sc4-1BBL or Fc−scCD70 or Fc−scGITRL (lower). Positive staining is shown by white histogram. Negative staining control is shown by shaded histogram. (**B**) IkBa degradation mediated by Fc−scTNFL proteins. T cell hybridoma cells were incubated with 30 µg/mL of Fc−scOX40L or Fc−sc4-1BBL or Fc−scCD70 or Fc−scGITRL for indicated periods to evaluate the expression of IkBa and b-actin proteins by immunoblotting. Graphs indicate the relative expression levels of IkBa, as determined by densitometry (IkBa/b-actin). (**C**) Increased expression of 4-1BB on T cell hybridoma cells after stimulation with PMA and Ionomycin. T cell hybridoma cells were cultured with 6.25 ng/mL PMA and 125 ng/mL Ionomycin for 8 h, followed by washing and resting for 3 h to detect the expression of 4-1BB by flow cytometry. (**D**) IkBa degradation mediated by Fc−sc4-1BBL in activated T cell hybridoma cells. T cell hybridoma cells activated with PMA/Ionomycin were prepared as in (**C**), and the level of IkBa was evaluated as in (**B**). Data are from one experiment representative of at least two independent experiments with similar results.

**Figure 6 cells-12-01596-f006:**
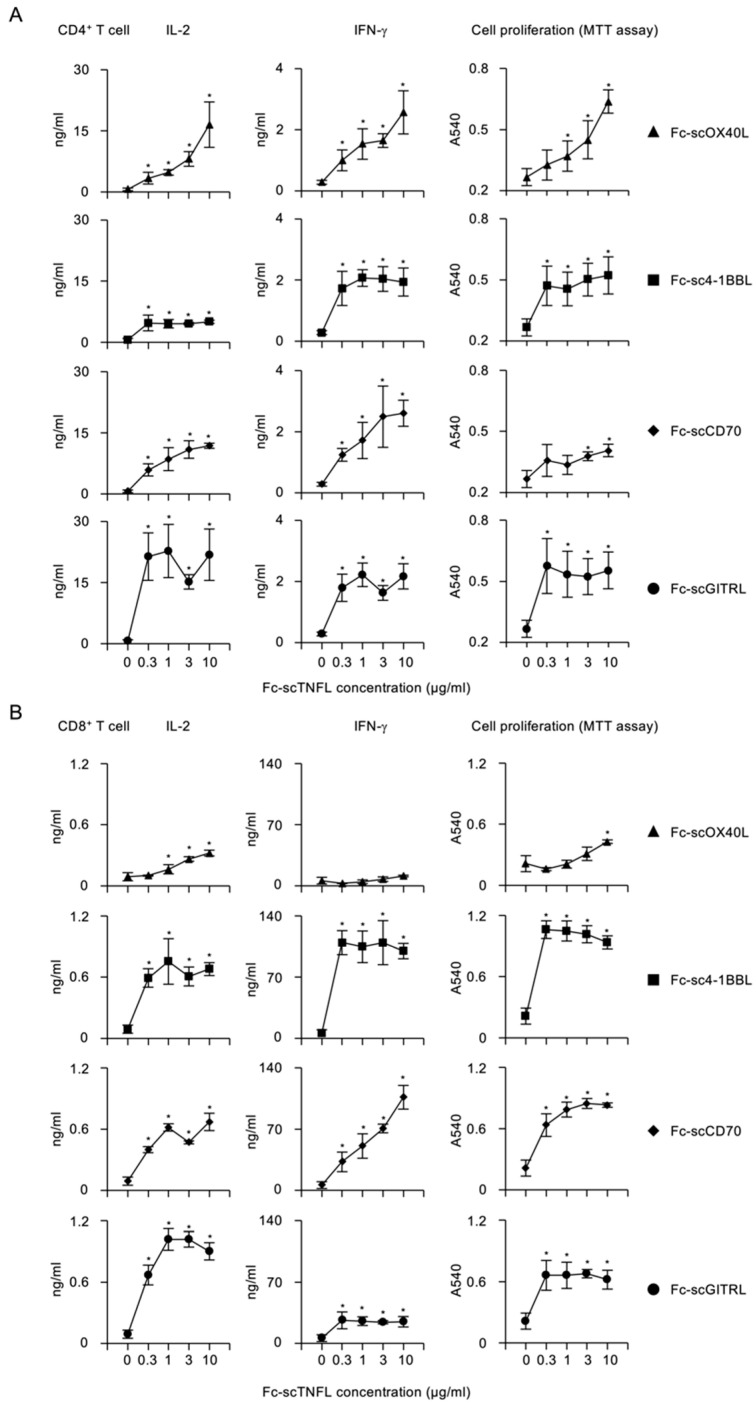
Dose-dependency of Fc−scTNFL proteins for the activation of CD4^+^ and CD8^+^ T cells. CD4^+^ (**A**) or CD8^+^ (**B**) T cells (5 × 10^4^ cells/well) were cultured in 96-well flat-bottomed plates precoated with 10 µg/mL anti-CD3 antibody in the presence or absence of indicated concentrations of soluble Fc−scOX40L or Fc−sc4-1BBL or Fc−scCD70 or Fc−scGITRL for 3 days. Cell proliferation was evaluated by MTT assay. Cytokine concentration was determined by ELISA. Data are mean ± standard deviation (*n* = 3) and from one experiment representative of at least two independent experiments with similar results. * *p* < 0.05 for comparison between with and without Fc-scTNFL in respective concentrations (Student *t* test).

**Figure 7 cells-12-01596-f007:**
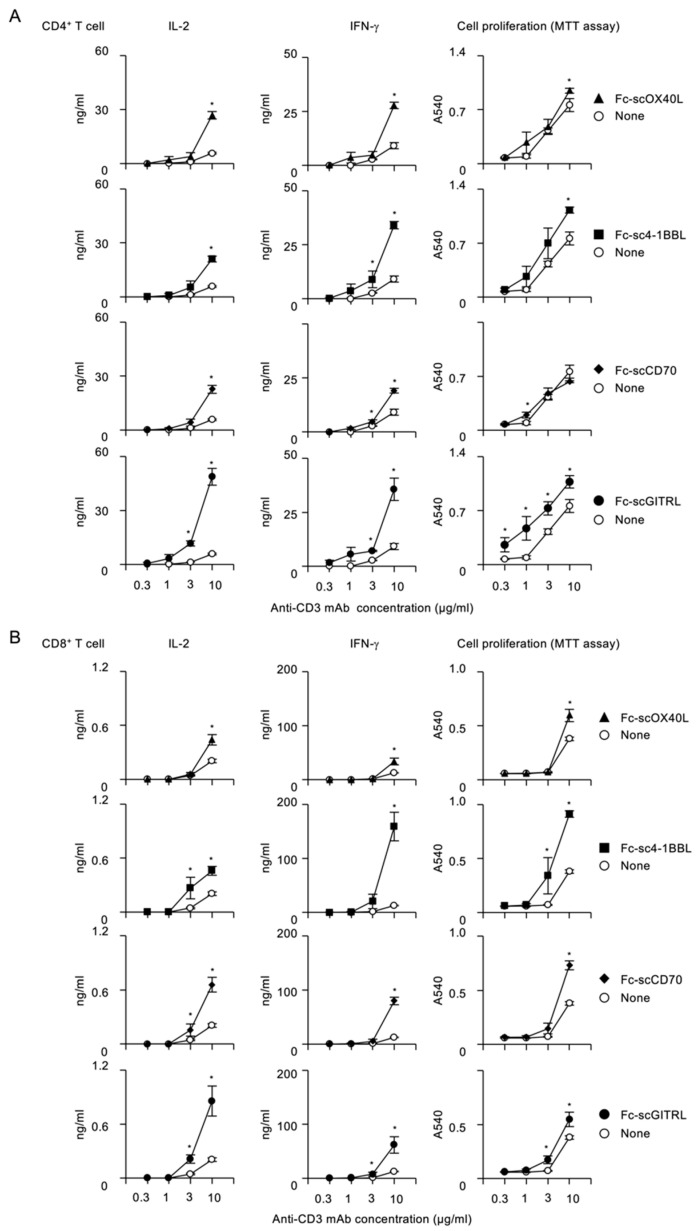
Requirement of higher concentrations of anti-CD3 for the activation of costimulatory signaling pathway mediated by Fc−scTNFL proteins. CD4^+^ (**A**) or CD8^+^ (**B**) T cells (5 × 10^4^ cells/well) were cultured in 96-well flat-bottomed plates precoated with indicated concentrations of anti-CD3 antibody in the absence (none) or presence of 1 µg/mL of respective Fc−scTNFL proteins for 3 days. Cell proliferation was evaluated by MTT assay. Cytokine concentration was determined by ELISA. Data are mean ± standard deviation (*n* = 3) and from one experiment representative of at least two independent experiments with similar results. * *p* < 0.05 for comparison between with and without Fc-scTNFL in respective anti-CD3 concentrations (Student *t* test).

**Figure 8 cells-12-01596-f008:**
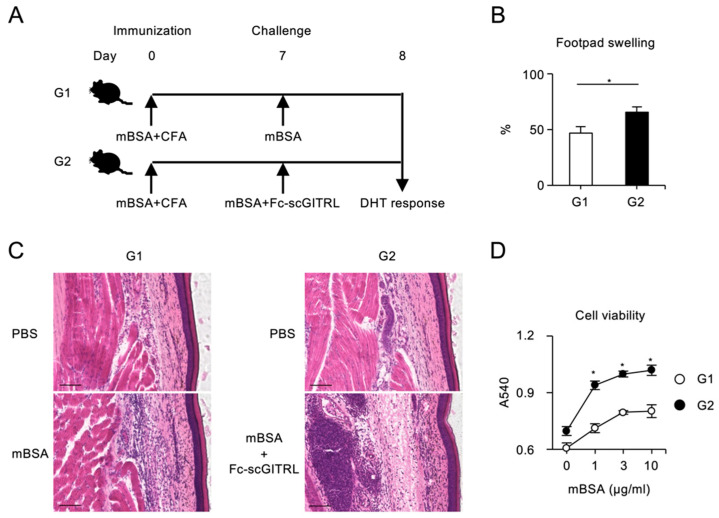
Enhanced DTH by challenge injection of Fc−scGITRL. (**A**) Schematic diagram of the experimental schedule for inducing DTH response, immunized with mBSA and CFA, and challenged with mBSA (210 µg/mouse) only (group 1, G1) or mBSA (210 µg/mouse) plus Fc−scGITRL (10 µg/mouse) (G2). (**B**) Footpad swelling on day 8. Data are average and standard error of the mean of seven mice per group. (**C**) Histological analysis of footpads on day 8. Histological sections from PBS or mBSA injected footpad in G1 (left) and PBS or mBSA plus Fc−scGITRL injected footpad in G2 (right) were stained with hematoxylin and eosin (H&E). Scale bar, 100 µm. (**D**) mBSA-specific T cell proliferative responses. Pooled spleen cells from mice immunized and challenged with mBSA only (G1) or mBSA plus Fc−scGITRL (G2) were cultured with indicated concentrations of mBSA for 3 d. Cell proliferation was assessed by MTT assay. The average and the standard error of the mean of three wells are shown. * *p* < 0.05 (Student *t* test).

## Data Availability

The data that support the findings of this study are available from the corresponding author T.S. upon reasonable request.

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
