# Peer review of "Fundamental Characterization of Antibody Fusion-Single-Chain TNF Recombinant Proteins Directed against Costimulatory TNF Receptors Expressed by T-Lymphocytes"

_cells, 2023, doi:10.3390/cells12121596_

Round 1

Reviewer 1 Report

CELLS 2023 FCTNF LIGANDS CO-STIMULATORY T CELLS Nagai et al.

The Author provide a demonstration that soluble Fc−scTNFL proteins can activate CD4+ and CD8+ T cells. They have focused on OX40L ; 4-1BBL ; CD70 and GITRL. The manuscript is well balanced, experiments are most of the time well performed and the content is easy to read. Conclusions appear to be correct but I would suggest a few amendements to streghten it. Read below :

#1 First of all, given that the whoel work relies on the production of TNFSF ligands, the vector should be presented. Author refer to two references to describe the vector. Yet it is unclear from these pubications which vector has been used. You MUST provide a clear description here and explain its backbone and principal elements that allow you to produce your proteins, but also whether this vector is commercially available or not and if so how can it be obtained, shoudl the reader like to have it.

#2 : Figure #1 Authors should provide theoretical Mw of expected protein product and explain in the text why was the Pa-peptide used for ? We guess it has been included in the constrcut here in rder to perform an Elisa but this might be useful to explain it right from here !

#3 Along the line, Authors should comment why the Elisa isn’t working for Fc-CD70 and Fc-GITRL ! Is this due to the fact that the PA motif is masked or because your coating with CD70 and GITR didn’t work ! What proof have the authors that their coating was correct ! Please comment and show proofs if possible ie using an anti-CD70 and anti-GITR. Has this been done ?

#4 Figure #8 controls are missing in both panels : anti-CD3 cells alone and anti-CD3/CD28 should be included in this figure. Alone the line, cell death should be quantified using an annexin V staining and proliferation shoudl be monitored using CFSE.

#5 Last but not least, the Authors have not compared TNFSF trimers (cross-linked or not) vs hexamers (the single chains Fc ligands produced here). Will there be any significant difference with respect to CD4 and/or CD8 costimulation ? I expected to see such an experiment here ! This comparison would add value and in particular highlight the rational for using single chains Fc ligands !  I would also have expected this to be discussed.

The English is fine. 

Author Response

Please see the attachment pdf file.

Reviewer 2 Report

The article is very interesting and the author conducted consistent and demonstrative experiments. This direction in the regulation of the cell response through the effect on the expression of membrane-bound receptors seems to be very relevant.

I would like to note a few points to improve the understanding of readers of this material:

1. All experiments were performed on soluble proteins synthesized by the authors. Whereas under in vivo conditions, the activation of these receptors occurs during intercellular contacts. And this model, in which there are no intercellular interactions, is very artificial and certainly makes a certain impact on the results obtained. Perhaps this should be discussed and described in an article.

2. FC-scTNFL binding activity to CD4+ and CD8+ cells was measured before stimulation and after culture with anti-CD3. However, it is known that in vitro cultivation itself, especially for several days, changes the phenotype and functional state of cells, especially in a pure culture of subpopulations, so it would be more correct to evaluate this parameter in an intact and stimulated culture of CD4CD8 cells.

3. Figures 3,4,5 do not indicate errors, and do not indicate N

4. It is somewhat unclear why the analysis for NF-kB was not also performed on CD4 and CD8 cells; this would greatly enrich the work.

5. In figure 6 below, the signature should apparently be Fc-scTNFL

6. Since all receptors had different expression levels, in particular, anti-CD3 stimulation led to an increase in the level of OX-4 expression and a decrease in CD27 expression, it would be extremely valuable to correlate expression levels with the response of cells

7. In the materials and methods, only Student's test is indicated as a statistical test, I would like to clarify whether the samples were tested for normal distribution.

8. Why, when assessing the degree of activation of the T-cell receptor using antiCD3, the concentration was titrated only downward: 0-10 μg

9. Perhaps Figure 8 would look more logical after Figure 7, since it is its logical continuation and a different presentation of the same data

10. For a number of receptors, including receptors for the mediator TNF, it has been shown that co-expression is critical. Whether the analysis of the level of simultaneous expression of the studied receptors was carried out.

Author Response

Please see the attachment pdf file.

Reviewer 3 Report

In this manuscript, authors try to tease out how certain co-stimulatory signals enhance T cell effector function. The production of these molecules is not trivial, and I would like to congratulate the authors for making functional molecules.

The premise of this manuscript is interesting. However, I do think that this manuscript needs a bit more work before it can be considered for publication. The figures are of extremely poor quality, information regarding the amount of replicates/experiments is missing in many parts of the results section, some of the data is interpreted as "significantly different" just because a statistical test says so while the results are most likely not of biological significance, and the discussion lacks in two places - how do the all-murine results translate to usable strategies for the treatment of malignancies in humans, and how these results impact the design of (novel) CAR-T modules. Unfortunately, this means I can only reccommend major revisions.

Major

1.       The figures are of really poor quality, it’s hard to read the axes sometimes. Please have a critical look and improve where possible.

2.       Please provide information regarding the representativeness / amount of replicate wells/animals/experiments/etc for figure 1, figure 2, figure 3, figure 4, figure 5

3.       Line 381 – I do not agree. For CD8s, OX40 does not do much. Authors add a star to the IL-2 results, but actually for all IL-2 measurements in the CD8 population, the amount of IL-2 is of no significance. An increase from 0.1 to 0.4 ng/mL (!) will probably not matter much in a biological setting. Please amend the conclusion of this figure to reflect this.

4.       Discussion – I am missing any reference to CAR-T cells, where these costimulatory domains are of prime importance. Authors need to add how their work could be exploited to enhance treatment strategies, or explain why some costimulatory domains are better than others. Furthermore, I am also missing how the all-mouse results translate to a human setting.

General

For all flow cytometry plots, please add back the numbers to the axes so readers can quanitify expression if they would like to. If in one figure (complete figure, or panel, whatever is appropriate) the axes are all similar for, let’s say, FITC (Fig 2D), it is sufficient to add the numbers to just one of the flow plots.

Specific

Line 14 – please refer to co-stimulatory signal, or co-stimulation (dash optional in both) as such, instead of cosignal, throughout the manuscript, as that is the more accepted way of referring to stimulatory signals that potentiate but require TCR-mediated T cell activation

Line 141 – please list the source of the HEK293T cells.

Flow cytometry and ELISA sections – please list the dilutions of all antibodies used.

Line 230 – please list age and sex of mice used.

Line 235 – please list concentration of anti-CD3 used.

In the results section (Figure 4), the legend speaks of anti-CD28 and IL-2 stimulation – this is not mentioned in the Materials section. Please add.

Figure 4 – please add headers above each column (CD40, 4-1BB, CD27, GITR), as the whole column refers to 1 protein. Similar to what authors did for figure 5. This will make it a bit more insightful for readers.

Line 365 – not necessarily true, T cells can also proliferate upon sensing solely cytokines or danger signals or TLR ligands. It however does go most efficient in the presence of the TCR. Please amend.

Line 373 – please use a gamma signal for IFN-g.

Figure 6 – cell viability is mentioned here (which is a bit more suitable for MTT assays), while the text speaks of proliferation. Please be consistent.

Figure 7 – I understand the point authors want to make, but first of all, the gigantic amount of TCR signalling shown here will never occur in vivo, especially not in oncological settings. I hope this comes in the discussion.

Figure 8 – this figure is redundant, as it recapitulates the same data as figure 7 at the 10 ug/mL condition. I recommend removal, unless authors come up with a very clear reason as to why to include this. Otherwise, I would propose merging Figures 7 and 8, if authors want to keep the summary panels (which I can imagine).

Line 370 – fix = fixed

Author Response

Please see the attachment pdf file.

We appreciate if the reviwer would consider two points below.

(1) The reviewer indicated typos, IFN-g. “g” should be Symbol font. This is found in the template file but not in our original text file in Microsoft Word. Please see the attached text.

(2) The reviewer indicated that the figures are poor quality. We think that the figures in pdf file attached should be fine, but if the quality is not enough , please let us know.

Round 2

Reviewer 1 Report

Thanks for addressing my comments.

Reviewer 2 Report

Thank you for your attention to comments.

Reviewer 3 Report

Thank you for taking my suggestions into account, no further comments.